# Research on aging behaviors of insulating silicone oil for cable terminals based on chromatographic and spectroscopic analysis

Wei Zhang[1]*, Jie Chen[1], Chenying Li[1], Jingying Cao[1], Xiao Tan[1], Chao Gao[2]

1 Electric Power Research Institute of State Grid Jiangsu Electric Power Co., Ltd., 2 State Grid Jiangsu Electric Power Co

* zhangweijsdl@163.com

## Abstract

With the widespread deployment of high-voltage cable terminals in power systems, insulating silicone oil has become a critical medium due to its superior dielectric and thermal properties. However, conventional diagnostic methods such as the three-ratio gas analysis developed for transformer oil have proven ineffective for silicone oil, owing to its distinct chemical structure and degradation behavior. To address this, this study aims to establish a fault-type identification method specifically for silicone oil to enhance the operational reliability of cable terminals. Accelerated thermal aging experiments (140°C, 30 days) were conducted to simulate long-term aging of silicone oil. By integrating partial discharge, high-energy discharge, and breakdown experiments, the gas generation patterns of silicone oil under different stresses were systematically analyzed. Gas chromatography (GC) and infrared spectroscopy (IR) were employed to track gas composition and chemical structural changes. The results propose the following diagnostic criteria: $H_2/CH_4 > 1$, $C_2H_4/C_2H_6 > 1$, and $C_2H_2/C_2H_4 < 0.1$ indicate overheating faults; $H_2/CH_4 < 1$, $C_2H_4/C_2H_6 < 0.1$, and $C_2H_2/C_2H_4 < 0.1$ correspond to partial discharge; while $H_2/CH_4 > 1$, $C_2H_4/C_2H_6 > 0.1$, and $C_2H_2/C_2H_4 > 5$ signify high-energy discharge. In addition, kinetic modeling based on the Arrhenius equation was applied to extract the activation energy of pyrolytic gas formation, confirming its relation to Si–O bond cleavage. This research provides a foundation for fault diagnosis in silicone oil-insulated equipment, effectively improving the operational reliability of power systems.

## 1. Introduction

Power cables, as critical components of modern urban power grids, are rapidly evolving and play an indispensable role in transmission and distribution systems. Their safe and reliable operation directly affects grid stability and holds strategic importance for ensuring continuous power supply, minimizing fault-related outages,

**Data availability statement:** All relevant data are within the paper and Supporting Information files or available at Mendeley Data (doi: 10.17632/yjpfcgm8v4.1).

**Funding:** This work is supported by Science and Technology Project of State Grid Jiangsu Electric Power Co., Ltd. (Research on Online Monitoring Technology for Multiple Parameters in Oil-Filled Cable Terminals, No. J2024043). The funder had no role in study design, data collection and analysis, decision to publish, or preparation of the manuscript.

**Competing interests:** The authors have declared that no competing interests exist.

and enhancing power quality. In particular, within power systems operating at 110 kV and above, cable terminals—key nodes linking cables to transformers and circuit breakers—serve not only as essential conduits for power flow but also face multiple challenges such as high voltage stress, temperature fluctuations, and mechanical forces. As a result, the insulation performance of these terminals is a decisive factor in the long-term stability of cable systems [1].

To meet the increasingly stringent insulation requirements of high-voltage cable terminals, oil-filled insulation structures have become widely adopted. Among various insulating media, silicone oil has emerged as a preferred choice due to its excellent thermal stability, chemical inertness, and superior dielectric properties. Its broad operational temperature range, from −50°C to 200°C, ensures fluidity even under extreme cold conditions. Moreover, silicone oil exhibits strong resistance to long-term thermal aging, effectively reducing the formation of deposits and sludge that commonly degrade insulation performance, thereby prolonging the service life of electrical equipment. Additionally, its low dielectric loss (tan δ < 0.001) and outstanding chemical stability maintain consistent electrical and mechanical protection under harsh environmental and electrical stresses [2–4].

However, studies have revealed significant degradation of silicone oil used in Gas Insulated Switchgear (GIS) terminals under partial discharge (PD) conditions. PD activity leads to a 30%−50% reduction in volume resistivity and a 20%−40% decrease in breakdown voltage. Visually, the oil undergoes discoloration from transparent to dark brown, reflecting considerable deterioration in insulation and physico-chemical properties. Experimental investigations into abnormal heating faults in 110 kV porcelain-insulated cable terminals have confirmed that increased dielectric loss (>0.005) in silicone oil results in thermal accumulation, triggering high-temperature alarms—primarily caused by inherent quality defects [5–7]. Comprehensive analyses of insulating oil selection criteria for high-voltage cable terminals highlight that developing scientific selection indices, accounting for climatic conditions, operating environments, and material compatibility, can optimize oil choice and improve terminal reliability.

Despite the growing application of silicone oil in electrical insulation systems, effective and validated diagnostic criteria for identifying fault types in silicone oil-based equipment are still lacking. Conventional fault diagnosis methods—such as the widely used three-ratio method based on gas chromatography data (e.g., $C_2H_2/C_2H_4$, $CH_4/H_2$, and $C_2H_4/C_2H_6$)—have been effective for transformer oils due to their well-understood decomposition behavior under thermal and electrical stresses [8–11]. However, applying these traditional diagnostic ratios to silicone oil often produces inconsistent or misleading results, as silicone oil differs fundamentally in chemical composition, oxidation mechanisms, and thermal degradation kinetics. Notably, silicone oil generates significantly lower acetylene ($C_2H_2$) levels and exhibits distinct gas evolution patterns during aging and discharge processes, making transformer oil-based criteria unsuitable. These differences emphasize the urgent need to develop silicone oil-specific fault classification methods. In this context, our research proposes and experimentally validates a novel set of gas ratio indicators tailored specifically

for silicone oil, aiming to address this diagnostic gap and enhance the accuracy and reliability of condition monitoring in high-voltage insulation systems.

## 2. Experimental methods

### 2.1. Sample

The silicone oil sample selected for testing was AK-50 insulating silicone oil produced by Wacker Chemie AG, Germany. Its primary component is polydimethylsiloxane (PDMS), featuring a linear chain structure composed of alternating silicon and oxygen atoms, with each silicon atom bonded to two methyl groups. The chemical structural formula is presented in Fig 1, and the molecular structural formula is presented in Fig 2. The key physical and chemical properties of the silicone oil sample are summarized in Table 1 (The technical datasheet provided by the manufacturer, Wacker Chemie AG).

The viscosity of silicone oil depends on its molecular weight, which is controlled by the number of dimethylsiloxane units(n). The higher the molecular weight, the greater the viscosity of the oil. For silicone oil with a viscosity of 50, the number of dimethylsiloxane units (n) is 40, and the molecular weight is 3000. Although carbon chains in organic

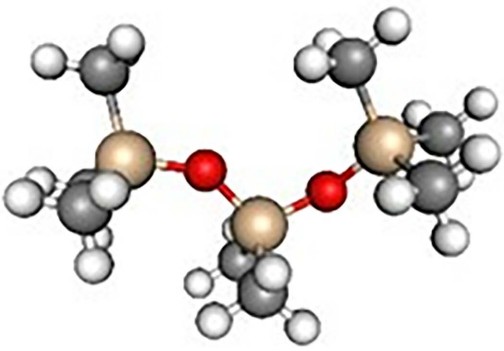

**Fig 1. Chemical structure of AK-50.**

**Fig 2. Molecular structure of AK-50.**

**Table 1. Basic parameters of test sample silicone oil.**

| Basic parameters | Performance index |
|---|---|
| Density (25°C) | 0.96 g/cm$^3$ |
| Kinematic Viscosity | 50 mm$^2$ s$^{-1}$ |
| Coefficient of thermal expansion at 0–150°C | 9.5 cm$^3$/ (cm$^3$°C)×10$^{-4}$ |
| Thermal conductivity at 50°C | 0.15 W K$^{-1}$m$^{-1}$ |
| Flash point | >250°C |
| Pour point | 55°C |
| Volatility | <2% |

compounds are susceptible to external influences, the Si—O bonds in inorganic compounds are typically as stable as inert silicates. Therefore, silicone oil exhibits stable chemical properties.

## 2.2. Silicone oil aging and testing platform device

In this study, silicone oil (AK-50) was selected as the research subject, and an accelerated thermal aging experiment was conducted in a constant-temperature oven at 140°C for 30 days. Experimental research was carried out on the oil chromatography and infrared spectroscopy of silicone oil in different aging states, as well as the oil chromatography of silicone oil under different discharge conditions [12]. During the long-term high-temperature aging process of 30 days, samples were taken every 6 hours, 1 day, 5 days, and subsequently every five days (Short-term intervals (6 h, 1 d) aim to capture rapid changes in the early stages of thermal decomposition, while the longer intervals (≥5 d) are used to monitor steady-state gas accumulation and long-term aging progression). Oil chromatography and infrared spectroscopy were performed on each sample. Oil chromatography mainly measures the concentrations of seven gas components produced during the aging process of silicone oil, namely hydrogen ($H_2$), methane ($CH_4$), ethane ($C_2H_6$), ethylene ($C_2H_4$), acetylene ($C_2H_2$), carbon monoxide ($CO$), and carbon dioxide ($CO_2$). The ratios of these seven gases are used to preliminarily determine the fault type of the silicone oil [13,14]. Infrared spectroscopy is used to measure the absorption peaks of silicone oil to analyze the stability of its chemical structure groups and to determine whether there is a relationship between the generated gases and the bond breakage. This provides theoretical support and data reference for the fault type judgment of silicone oil used in high-voltage cable terminals. The silicone oil is sealed in a clean bottle and placed in a drying oven for high-temperature thermal aging. Fig 3 and Fig 4 are photographs of the experimental oven and the silicone oil, respectively, under different aging states.

The partial discharge measurement of silicone oil was conducted using a customized needle-plate electrode. The experimental container was made of glass, cylindrical in shape, with good transparency and insulation properties, facilitating observation of the experimental process and ensuring experimental safety. The needle-plate electrode consists of a needle electrode and a plate electrode, with a gap distance of 2.5 mm between them. The plate electrode, made of brass, has a cylindrical base with a diameter of 30 mm to ensure stability and conductivity, simulating the partial discharge conditions in actual working conditions. Two leads were extended from the needle-plate electrode to connect to the high-voltage power supply and detection equipment to ensure accurate voltage application and signal detection. Fig 5 shows the schematic diagram of the experimental circuit. Fig 6 presents a physical photograph of the needle-plate electrode. Voltage was slowly applied at a rate of 1 kV/s until the initial discharge signal was detected in the oil sample. After detecting the

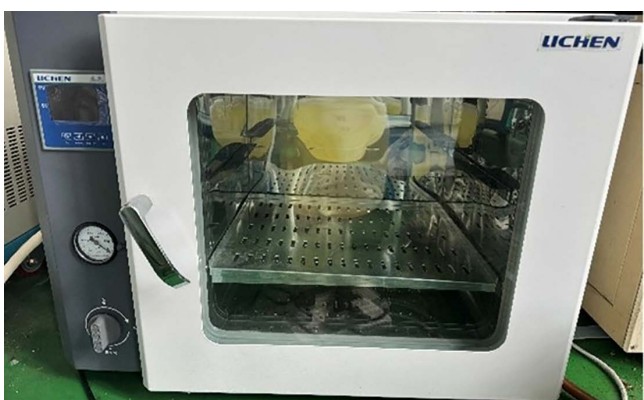

**Fig 3. Actual photograph of the oven.**

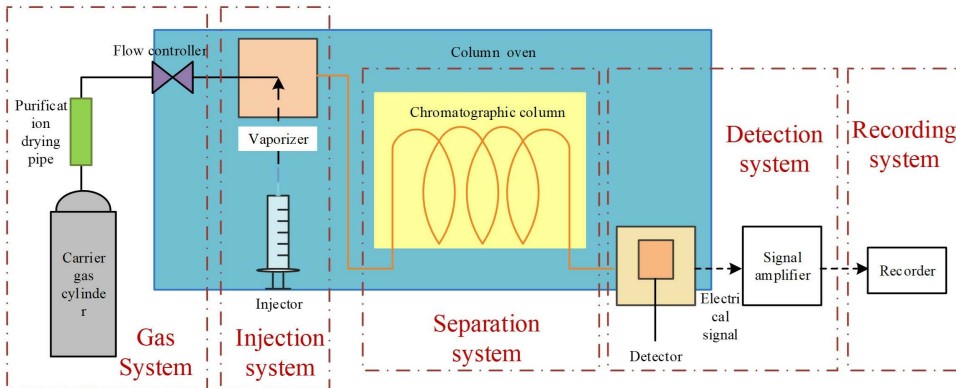

**Fig 4. Silicone oil at different aging stages.**

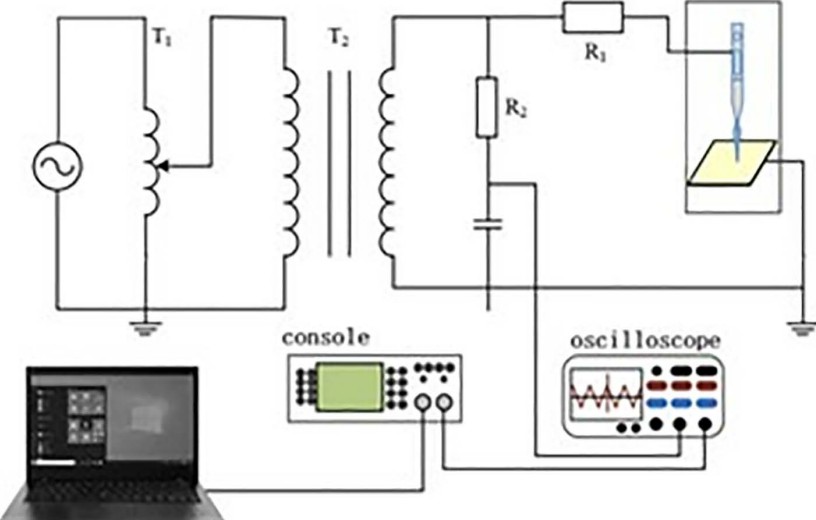

**Fig 5. Partial discharge experimental circuit.**

discharge signal, the current voltage was maintained and the discharge continued for half an hour. After the discharge was completed, samples were taken from the container for oil chromatography analysis to examine the changes in gas components. The silicone oil in the container was replaced with fresh oil, and the voltage application and discharge process were repeated to ensure consistent experimental conditions. Each set of experiments was repeated three times to ensure the reliability and consistency of the results.

Through oil chromatography analysis, the changes in gas components of silicone oil under different discharge conditions were examined to determine the fault types of silicone oil under partial discharge conditions.

Oil chromatography analysis primarily detects seven gas components generated during aging degradation: $H_2$: Reflects moisture or hydrocarbon decomposition in insulation materials. $CH_4$, $C_2H_6$, $C_2H_4$, $C_2H_2$: Characterize hydrocarbon pyrolysis or corona discharge. $CO$, $CO_2$: Indicate decomposition of oxygen-containing groups in silicone oil. $C_2H_2$: A signature gas for high-energy discharges. Considering the instrument's single measurement requirement of 40 mL, extract 60 mL

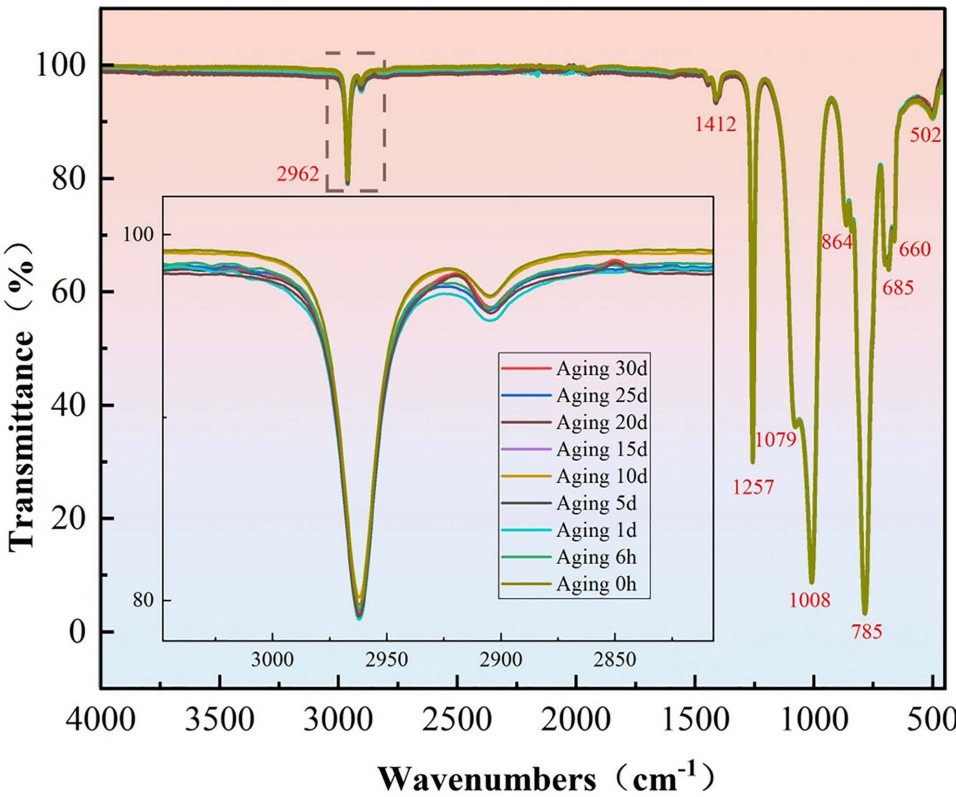

**Fig 6. Physical drawing of needle plate electrode.**

of post-discharge silicone oil sample. Then inject 1 mL of high-purity nitrogen as a carrier gas to assist gas separation and avoid oxygen interference. Place the sample in a constant-temperature shaker at 50°C for 20 minutes to promote gas liberation. After shaking, use a gas-tight syringe to extract gas from the sample headspace, avoiding external air contamination, and immediately proceed to measurement. The schematic flow diagram illustrating the principle of oil chromatography detection is shown in Fig 7. A chromatographic column selectively adsorbs gases based on polarity. Low-molecular-weight gases ($H_2$, $CH_4$) with weak adsorption elute first. Polar gases (CO, $CO_2$) interact strongly with the stationary phase, prolonging retention time. Hydrocarbon gases ($C_2H_6$, $C_2H_4$, $C_2H_2$) are separated sequentially by carbon chain length and double bond count. The thermal conductivity detector is used for detecting flameless combustion gases such as $H_2$, CO, and $CO_2$. Quantifies hydrocarbons ($CH_4$, $C_2H_6$) by ionizing them in a hydrogen flame and measuring the resulting current. Gas species are identified by comparing peak retention times with standard gas calibrations. Concentrations are calculated using the internal standard method, correlating peak areas with calibration curves.

By tracking the types and concentrations of characteristic gases, oil chromatography enables precise evaluation of silicone oil insulation degradation, providing critical data for fault early warning and lifespan prediction in electrical equipment (e.g., transformers, cables) [15].

Infrared spectroscopy detection is a method based on the characteristic absorption of infrared light by substances to analyze molecular structure and chemical composition. By conducting infrared spectroscopy on different samples and analyzing the wavelength positions and intensities of the absorption peaks, specific chemical bonds or functional groups in silicone oil can be identified. By comparing the infrared spectra of silicone oil at different aging times, the presence of

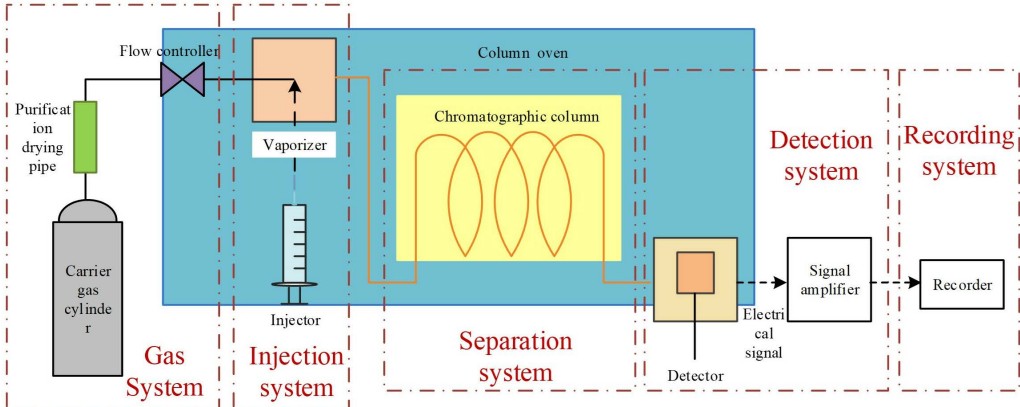

**Fig 7. Schematic diagram of oil chromatography testing device.**

bond breakage in the silicone oil can be observed to assess its thermal stability. Meanwhile, the results obtained from the detection can be used to determine whether there is a relationship between the amount of gas generated and the bond breakage [16,17].

After the experimental samples were placed into the test bottles, they were put into the drying oven for vacuum extraction. Once the pre-treatment of the experimental samples was completed, all samples were sealed and placed into the aging chamber for accelerated thermal aging experiments. Samples were taken at different aging times, with sampling times set at 0h, 6h, 1d, 5d, 10d, 15d, 20d, 25d, and 30d. After sampling, the samples were immediately sent to the testing site for oil chromatography measurements. The figure below shows the oil chromatography results obtained from the continuous aging process.

## 3. Results and analysis

### 3.1. Oil chromatography

**3.1.1. High-temperature oil chromatography.** The oil chromatography data detected during continuous 30-day thermal aging are presented in Fig 8 and Fig 9. Under high-temperature conditions, silicone oil undergoes pyrolysis reactions, primarily generating $CO_2$ and CO, with minor amounts of small-molecule hydrocarbons (such as $CH_4$, $C_2H_6$, and $C_2H_4$), no $C_2H_2$ was produced. The generation rates of these gases significantly increase after 5 days of aging, indicating that the aging reaction may enter an acceleration phase.

Specifically, the concentration of $H_2$ suddenly rises after 5 days of aging, increasing from 80.77 ppm at 10 days to 323.3 ppm at 30 days. The concentration of CO shows a marked upward trend, rising from an initial 89.76 ppm to 1952 ppm at 30 days. The increase in CO typically indicates thermal aging or thermal faults within the oil. The content of $CO_2$ also increases significantly, reaching 6220 ppm at 30 days, which may suggest overheating or localized overheating phenomena. The concentration of $CH_4$ gradually rises, indicating that the equipment may have experienced overheating. The concentration of $C_2H_4$ increases from 0.12 ppm to 10.87 ppm, which is usually associated with high-temperature overheating faults and requires attention. The concentration of $C_2H_6$ remains low, indicating no severe thermal faults. The concentration of $C_2H_2$ is very low (with a maximum of only 8.4 ppm), indicating that no high-energy arcing faults have occurred.

Assuming that the thermal decomposition of silicone oil to produce gases follows the Arrhenius equation, we intend to use the Arrhenius equation to estimate the activation energy of silicone oil pyrolysis, in order to determine whether the gas generation is related to the breaking of chemical bonds. Equation 1 and 2 are the first-order kinetic reaction formula, and equation 3 is Arrhenius formula. The formation of gases follows first-order reaction kinetics:

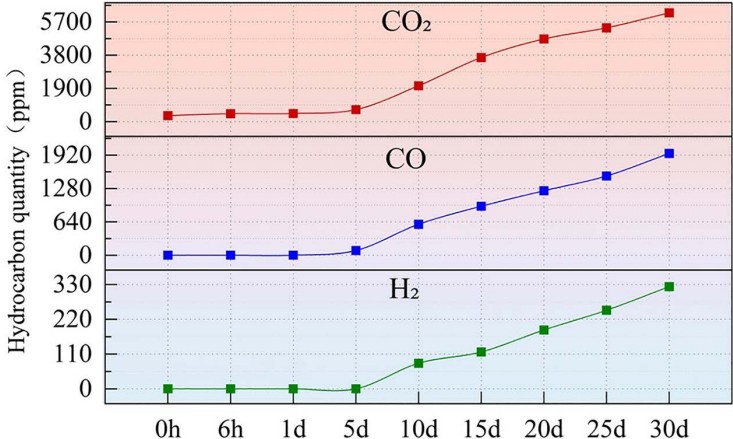

**Fig 8. Oil chromatography data for continuous aging over 30 days for $CO_2$, CO, $H_2$.**

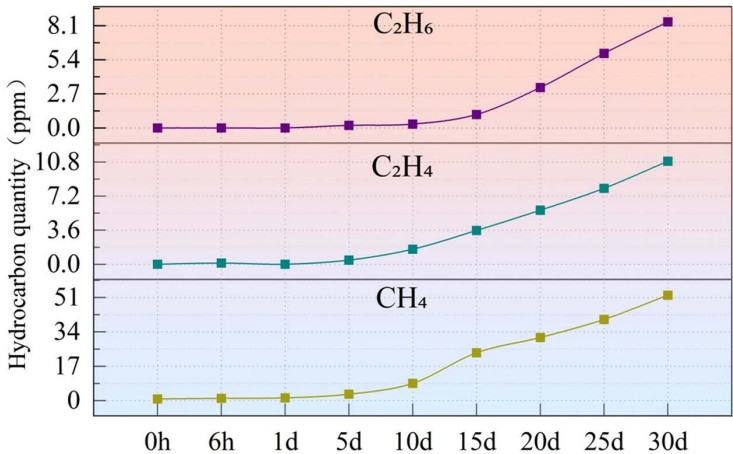

**Fig 9. Oil chromatography data for continuous aging over 30 days for $C_2H_6$, $C_2H_4$, $CH_4$.**

$$\frac{dC}{dt} = k \times (C_{max} - C) \tag{1}$$

$$\ln\left(\frac{C_{max}}{C_{max} - C}\right) = kt \tag{2}$$

Here, $C_{max}$ represents the maximum possible concentration, which is determined through fitting of experimental data, and k is the rate reaction constant.

Taking $H_2$ as an example, The example calculation for $H_2$ is presented in Table 2:

Arrhenius equation:

$$k = A \times e^{-\frac{E_a}{RT}} \tag{3}$$

**Table 2. Example Calculation for H$_2$.**

| Time (days) | H$_2$ Concentration (ppm) | $\ln\left(\frac{C_{max}}{C_{max}-C}\right)$ |
|---|---|---|
| 5 | 80.77 | 0.12 |
| 10 | 150.2 | 0.32 |
| 30 | 323.3 | 0.87 |

Assuming $C_{max}$ = 350 ppm, The fitting result is $k \approx 0.029$ day$^{-1}$.

Here, $R$=8.314, $T$=413K (experiment conducted at 140°C), assuming $A = 10^{12} day^{-1}$ Calculated using the above equation:

$$E_a = -R \times T \times \ln\left(\frac{k}{A}\right) = -8.314 \times 413 \times \ln\left(\frac{0.029}{10^{12}}\right) = 85\text{kJ/mol}$$

The bond energy of the Si-O bond is approximately 452 kJ/mol, and the bond energy of the C-H bond is approximately 413 kJ/mol. The activation energy $E_a$ is about 85 kJ/mol, which is significantly lower than the bond energy of the Si-O bond. This suggests that the reaction is unlikely to involve the scission of the main chain. The value is closer to the bond energy of the C-H bond, supporting the idea that the formation of H$_2$ and CO originates from the oxidation or cleavage of side chains (such as -CH$_3$), rather than the scission of Si-O bonds. The bond energies of other gases calculated using the aforementioned equations are presented in Table 3:

### 3.1.2. Partial discharge oil chromatography.
Partial discharge tests were conducted using needle-plate electrodes, with two different discharge scenarios.

Initial Discharge: After detecting a discharge signal on the oscilloscope, the voltage was maintained at the current level, and the discharge was continued for 30 minutes.

Intense Discharge: Based on the initial discharge voltage, the voltage was further increased without causing electrode breakdown, resulting in intense discharge, which was also sustained for 30 minutes. Each of the two sets of experiments was repeated three times. The oil chromatography data for different discharge stages are presented in Table 4.

From the data obtained, it is evident that the gas content generated during intense discharge is higher than that during initial discharge. However, during the partial dis-charge process, CO and C$_2$H$_2$ were not detected. The likely reason is that although the energy increased with the discharge stage, it did not reach the level required for sili-cone oil to produce

**Table 3. Calculation and Analysis of Chemical Bond Energy.**

| Gas | Activation Energy ($E_a$) | Bond dissociation energy (kJ/mol) | Mechanistic inference of the reaction |
|---|---|---|---|
| CH$_4$ | 78 kJ/mol | C-H: 413 | Side-chain physical volatilization |
| C$_2$H$_4$ | 72 kJ/mol | C=C: 614 | Free radical recombination |
| C$_2$H$_6$ | 85 kJ/mol | C-C: 347 | Possibly limited by diffusion |

**Table 4. Discharge oil chromatogram data.**

| Type/Gas | H$_2$ | CO | CO$_2$ | CH$_4$ | C$_2$H$_4$ | C$_2$H$_6$ | C$_2$H$_2$ |
|---|---|---|---|---|---|---|---|
| Initial Discharge | 0.87 | 0 | 648.54 | 1.15 | 0 | 0.29 | 0 |
| | 0.85 | 0 | 634.69 | 1.09 | 0 | 0.31 | 0 |
| | 0.88 | 0 | 645.32 | 1.12 | 0 | 0.28 | 0 |
| Intense Discharge | 0.94 | 0 | 681.58 | 1.42 | 0.04 | 0.47 | 0 |
| | 0.96 | 0 | 690.56 | 1.45 | 0.05 | 0.48 | 0 |
| | 0.93 | 0 | 685.34 | 1.44 | 0.04 | 0.47 | 0 |

an arc or undergo high-temperature decomposition. Therefore, $C_2H_2$ was not detected. Meanwhile, the detection of trace amounts of $C_2H_4$ and a higher concentration of $CO_2$ during intense discharge indicates that the discharge has begun to cause insulation aging in the silicone oil and may have led to localized overheating internally.

### 3.1.3. Breakdown oil chromatograph.

The breakdown oil chromatography was also conducted using a needle-plate electrode. For new samples, the voltage was gradually increased at a rate of 1 kV/s until the electrodes were punctured. After the breakdown, samples of the silicone oil were taken and subjected to oil chromatography analysis. The experiment was repeated three times. The oil chromatography data after breakdown are presented in Table 5.

From the data obtained, the content of $H_2$ has significantly increased compared to the values measured during discharge. The increase in $H_2$ content indicates that spark discharge has occurred in the oil. The content of $CO_2$ showed no significant change compared to the data measured during discharge. The contents of $CH_4$, $C_2H_4$, and $C_2H_6$ have all increased significantly compared to the previous experimental data, indicating that the silicone oil has undergone intense discharge. After breakdown, the oil chromatography detected CO and $C_2H_2$, which are quite different from the data measured during partial discharge. This also indicates that related fault types have occurred within the oil. The presence of $C_2H_2$ clearly points to the occurrence of high-temperature arcing.

## 3.2. Infrared spectroscopy

Fig 10 presents the infrared spectrum of AK50 silicone oil and the corresponding spectra after continuous aging for up to 30 days. Infrared spectroscopy was conducted to investigate whether aging induced chemical bond breakage and to explore any correlation between gas content changes and bond energy breakages. The results show no significant chemical bond breakage after 30 days of aging, indicating that the chemical structure of AK50 silicone oil remains highly stable. This suggests that gas generation during aging is not related to bond breakage but rather to thermal reactions leading to the decomposition of certain gases. A decrease in transmittance is also observed, which is attributed to the formation of small-molecule gases or volatile substances from thermal decomposition. These products can absorb infrared light and enhance scattering, thereby reducing transmittance.

As can be seen from the figure above, the main absorption peaks of the silicone oil are listed in Table 6, along with the analysis of the corresponding functional groups.

In summary, regarding the thermal aging of silicone oil, the results of the measurements, corroborated by the Arrhenius equation, indicate that the bond energy of $H_2$ is compared with the bond energies of Si-O and C-H bonds. The results suggest that the reaction does not involve the scission of the main chain. The generation of $H_2$ and CO is supported to originate from the oxidation or cleavage of side chains (such as $-CH_3$), rather than the scission of Si-O bonds. This conclusion is further reinforced by infrared spectroscopy data, which show no significant changes in the absorption peaks of Si-O (1079 $cm^{-1}$) or Si-C (785 $cm^{-1}$), thereby ruling out the possibility of main-chain scission. The generation of gases is thus decoupled from the stability of chemical bonds, consistent with the hypothesis of physical volatilization or side-chain decomposition.

Regarding partial discharge, low-energy discharges lead to surface oxidation of silicone oil (partial scission of Si-O bonds), resulting in the formation n of $CO_2$ and a small amount of $H_2$, but without reaching the energy threshold required to cleave C-H bonds. High-energy discharges and breakdowns, which produce arcs, generate high temperatures that trigger the cleavage of C-H bonds (leading to the formation of $C_2H_2$).

**Table 5. Breakdown oil chromatogram data.**

| Type/Gas | $H_2$ | CO | $CO_2$ | $CH_4$ | $C_2H_4$ | $C_2H_6$ | $C_2H_2$ |
|---|---|---|---|---|---|---|---|
| Breakdown | 8.35 | 1.65 | 667.54 | 2.33 | 0.4 | 1.37 | 3.18 |
| | 8.28 | 1.62 | 672.34 | 2.38 | 0.38 | 1.35 | 3.16 |
| | 8.32 | 1.64 | 670.26 | 2.32 | 0.41 | 1.38 | 3.22 |

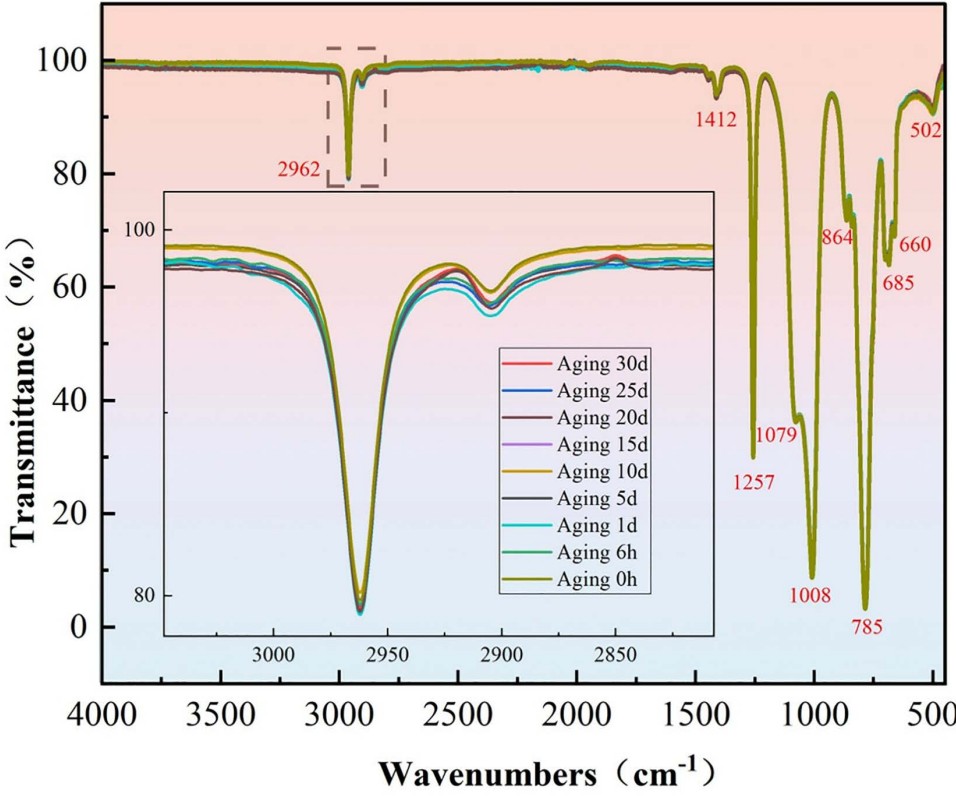

**Fig 10. Infrared spectrum of AK50 silicone oil and aging data over 30 days.**

**Table 6. Summary of main FTIR absorption peaks and functional group.**

| Wavenumber (cm⁻¹) | Vibrational Mode | Functional Group/ Source | Interpretation/ Structural Implication |
|---|---|---|---|
| 2962 | C–H stretching | $CH_3$, $CH_2$ (alkanes) | Indicates saturated hydrocarbon content |
| 1412 | C–H bending | $CH_3$, $CH_2$ (alkanes) | Typical of methyl/methylene groups |
| 1257 | C–O–C stretching | Ester, ether, C–O in polymers | Suggests presence of oxygen-containing polymer structures |
| 1079 | Si–O–Si stretching | Siloxanes, $SiO_2$ | Characteristic peak of silicone backbone |
| 1008 | C–O stretching | Ether or alcohol groups | Indicates oxidative structural modification |
| 864 | Si–OH bending | Silanol (Si–OH) | Reflects hydrolysis or aging-induced silanol formation |
| 785 | Si–C stretching | Organosilicon compounds | Characteristic of organic–inorganic hybrid linkages |
| 685, 660 | Si–OH bending (double peak) | Siloxanes, silicone rubber | Common in aged or moisture-exposed silicone structures |
| 502 | Si–O bending (deformation) | Si–O network | Low-frequency bending of siloxane framework |

Through a 30-day accelerated thermal aging experiment and partial discharge experiments of varying degrees, and by analyzing the results obtained, it has been decided to use the ratios of $H_2/CH_4$, $C_2H_4/C_2H_6$, and $C_2H_2/C_2H_4$ to preliminarily determine the fault type of silicone oil. The $H_2/CH_4$ ratio is a key indicator reflecting the degree of material pyrolysis. When $H_2/CH_4 > 1$, it indicates that the temperature has exceeded the pyrolysis threshold of silicone oil, and the methylsiloxane chain has broken to generate free hydrogen. The $C_2H_4/C_2H_6$ ratio is a dynamic indicator characterizing the duration of thermal stress. A ratio > 1 indicates persistent overheating, with the generation rate of ethylene exceeding that of ethane.

The $C_2H_2/C_2H_4$ ratio is a sensitive parameter reflecting the discharge energy density. During high-energy discharge, the production of acetylene increases sharply.

Based on the experimental data, the following conclusions can be drawn: For silicone oil, if the gas content satisfies the condition $H_2/CH_4 > 1$, $C_2H_4/C_2H_6 > 1$, and $C_2H_2/C_2H_4 < 0.1$, it can be considered as an overheating fault. If the gas content satisfies $H_2/CH_4 < 1$, $C_2H_4/C_2H_6 < 0.1$, and $C_2H_2/C_2H_4 < 0.1$, it can be identified as partial discharge. Additionally, if the gas content satisfies $H_2/CH_4 > 1$, $C_2H_4/C_2H_6 > 0.1$, and $C_2H_2/C_2H_4 > 5$, it indicates high-energy discharge. These diagnostic criteria represent a significant advancement over traditional transformer oil fault diagnosis methods, which commonly use the three-ratio method ($C_2H_2/C_2H_4$, $CH_4/H_2$, and $C_2H_4/C_2H_6$) to identify fault types. However, these ratios, while effective for transformer oil, do not yield reliable results when applied to silicone oil due to its distinct chemical structure and decomposition behavior. For silicone oil, the relative production of gases such as acetylene ($C_2H_2$) is much lower, and the gas evolution patterns are fundamentally different. This study provides a set of gas-ratio indicators specifically tailored for silicone oil, filling the gap in existing fault diagnosis methodologies and offering a foundation for more accurate fault identification in silicone oil-insulated equipment.

## 4. Conclusions

This study investigates the thermal aging behavior of insulating silicone oil in high-voltage oil-filled cable terminations. Accelerated thermal aging experiments revealed that the silicone oil gradually decomposes with increasing aging time, accompanied by a continuous rise in gas content. However, Fourier-transform infrared spectroscopy (FTIR) showed no detectable bond cleavage, indicating the excellent thermal stability of the silicone oil. Based on gas chromatography analysis, we established a set of diagnostic criteria for fault classification. Specifically, when the gas content satisfies $H_2/CH_4 > 1$, $C_2H_4/C_2H_6 > 1$, and $C_2H_2/C_2H_4 < 0.1$, the condition is identified as overheating. A profile with $H_2/CH_4 < 1$, $C_2H_4/C_2H_6 < 0.1$, and $C_2H_2/C_2H_4 < 0.1$ corresponds to partial discharge, while $H_2/CH_4 > 1$, $C_2H_4/C_2H_6 > 0.1$, and $C_2H_2/C_2H_4 > 5$ indicate high-energy discharge. In summary, this work fills a critical gap in diagnostic techniques for silicone oil-insulated power equipment by proposing, for the first time, fault classification criteria based on GC-detectable gas ratios. These findings serve as a practical reference for the development of predictive maintenance and fault monitoring systems tailored to silicone oil applications.

## Supporting information

**S1 Data.  Article database.**
(XLSX)

## Author contributions

**Writing – original draft:** wei zhang, Jie Chen, Chenying Li, Jingying Cao, Xiao Tan, Chao Gao.

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
