## [Decision Letter · Decision Letter 0]

4 May 2025

Dear Dr. zhang,

We look forward to receiving your revised manuscript.

Kind regards,

Makungu Marco Madirisha

Academic Editor

PLOS ONE

Journal Requirements:

 [State Grid Jiangsu Electric Power Co., Ltd. science and technology project (Project No.: j2024043) "Research on multi parameter online monitoring technology of cable oil terminal"].

6. PLOS requires an ORCID iD for the corresponding author in Editorial Manager on papers submitted after December 6th, 2016. Please ensure that you have an ORCID iD and that it is validated in Editorial Manager. To do this, go to ‘Update my Information’ (in the upper left-hand corner of the main menu), and click on the Fetch/Validate link next to the ORCID field. This will take you to the ORCID site and allow you to create a new iD or authenticate a pre-existing iD in Editorial Manager.

Additional Editor Comments (if provided):

Reviewers' comments:

Reviewer's Responses to Questions

**Comments to the Author**

1. Is the manuscript technically sound, and do the data support the conclusions?

Reviewer #1: Yes

Reviewer #2: Partly

2. Has the statistical analysis been performed appropriately and rigorously?

Reviewer #1: No

Reviewer #2: No

3. Have the authors made all data underlying the findings in their manuscript fully available?

Reviewer #1: Yes

Reviewer #2: Yes

4. Is the manuscript presented in an intelligible fashion and written in standard English?

Reviewer #1: No

Reviewer #2: No

Reviewer #1: Research studies on aging behaviours of insulating silicon oil for cable terminals based on chromatographic analysis are essential to establish operational quality control measures. In this study, the authors have employed Gas Chromatography (GC) and Fourier Transform Infrared (FTIR) Spectroscopy to examine the aging characteristics of insulating silicon oil and established some criteria for fault diagnosis of insulating silicon oil. The study is meaningful; however, some issues need to be addressed before its publication:

1. The write-up of this manuscript doesn’t sound scientifically, the authors should follow proper scientific writing methods.

2. The title of the manuscript is not clear, consider revising it. The authors may consider a title such as “Research study on aging behaviours of insulating silicon oil for cable terminals based on chromatographic analysis” or improve it better.

3. The manuscript has some grammar and typo mistakes that need to be improved or corrected. For example, 140℃, 350ppm, etc. There should be a space between a number and a unit like 140 ℃ and 350 ppm. Also, in some areas there is no space after a period or full stop like “discharges.Considering” (line 132), and “interference.Place” (line 134). Make the necessary corrections throughout your manuscript.

4. The manuscript lacks consistency, for example, “Fig. 1.” and “Fig 2.” are used interchangeably. The authors should observe consistency throughout the manuscript.

5. In section 2 “Experimental and simulation methods”, which simulation method was used in this work and for what purpose? The ball and stick structure of silicon in Figure 1 is not clear, redraw and make it colourful.

6. The basic parameters given in Table 1 were they determined by the authors or other researchers? If authors, explain how they are determined or derived, and if other researchers, a reference is required.

7. Why the aging characteristics of the insulating silicon oil are tested only at 140 ℃ instead of a range of temperatures?

8. Figures 2 to 7, Tables 2 to 5, and Equations 1 to 3 are not mentioned or cited anywhere in the text. Figures, Tables, and Equations should be part of the text.

9. Why it is worthy reporting Figures 3 and 4 in the manuscript?

10. What is the unit for the y-axis in Figure 5? The y-axis title text direction should be changed.

11. Figures resolution is not good, consider redrawing them and make attractive.

12. Equations 1 to 3 are confusing; they should be properly written.

13. Generally, the manuscript in its current version lacks sufficient novelty. The author(s) should revise or reformulate the abstract, introduction, methodology, and conclusion to clearly show the strength and novelty of the manuscript.

Reviewer #2: The paper presents a timely and relevant study focusing on the diagnostic limitations of current methods for silicone oil in high-voltage cable terminals, however for improvement the following is recommended

Abstract.

1. Although the study’s contribution is clear, the abstract could more explicitly highlight how this work advances the state of the art over existing diagnostic methods.

2. The nature of the discharge tests (voltage levels, duration, etc.) could be briefly summarized to enhance context.

Introduction

1. Please avoid making general citations; instead, provide references that directly support specific claims. For example, lines 32 to 48 contain several assertions that lack citations. Be sure to support these arguments with evidence from relevant previous studies

2. Lines 51 to 54 lack citations

3. Avoid using numerical listings in the introduction unnecessarily. Instead, present the information in well-structured paragraphs to maintain a smooth and academic flow.

4. When building the case for the novelty of your study, it is important to clearly identify the limitations or gaps in previous research. Support this with specific examples and explain in detail how your study addresses these shortcomings. For instance, in the final paragraph lines (61 to 67), you stated: 'This research aims to establish scientific criteria for silicone oil-insulated equipment, thereby enhancing power system operational reliability.' However, the scientific criteria being proposed are not summarized or explained clearly. To strengthen your argument, briefly outline these criteria early in the discussion so the reader can immediately understand the specific contribution and significance of your work.

Experimental and simulation methods

1. In subsections 2.1 and 2.2, there are several instances of incorrect word breaks or hyphenation errors that affect readability and clarity. For example, 'ox-ygen atoms' should be corrected to 'oxygen atoms', 'ac-celerated' to 'accelerated', 'pro-cess' to 'process', and 'sin-gle-measurement' to 'single measurement'. Please carefully proofread all of subsections 2.1 and 2.2 to eliminate such typographical issues and ensure smooth and professional language throughout.

2. In lines 90 to 91, you mention: 'During the long-term high-temperature aging process of 30 days, samples were taken every 6 hours, 1 day, 5 days, and subsequently every five days.' However, the rationale for selecting these specific sampling intervals is not explained. Please clarify the basis for choosing these time points whether they are based on previous studies, preliminary tests, or specific degradation behavior expected in silicone oil. Providing justification will help readers understand the experimental design and its relevance to the study’s objectives.

Results and discussion

1. After reviewing the Results and Discussion sections, it is evident that your arguments are not sufficiently supported by references to previous studies. While the Introduction does mention some weaknesses in existing research, this context is not carried through into the discussion of your findings. This weakens the overall strength of your claims. I strongly recommend revisiting the Results and Discussion sections to compare your findings with those of previous studies. Supporting your interpretations with established literature will help validate your conclusions and clearly demonstrate the contribution and novelty of your work.

2. When summarizing detailed results such as those from FTIR analysis, consider presenting the key findings in a table for improved clarity and readability. A well-organized table can help highlight the main spectral changes, functional group variations, and trends across conditions more effectively than text alone, making it easier for readers to grasp and compare the results

3. Equations are not well presented example equations 1 to 3

Conclusion

Please refrain from using numerals to summarize your work; instead, present your findings in a well-organized paragraph. Your conclusion can also reflect your thoughts on the broader field and highlight the contributions your study makes to this area of expertise.

**Do you want your identity to be public for this peer review?** For information about this choice, including consent withdrawal, please see our Privacy Policy

Reviewer #1: No

Reviewer #2: **Yes: ** Henry Kahimbi

---

## [Author Response · Author response to Decision Letter 1]

3 Jun 2025

Response to Reviewer #1:

General comments:

Research studies on aging behaviors of insulating silicon oil for cable terminals based on chromatographic analysis are essential to establish operational quality control measures. In this study, the authors have employed Gas Chromatography (GC) and Fourier Transform Infrared (FTIR) Spectroscopy to examine the aging characteristics of insulating silicon oil and established some criteria for fault diagnosis of insulating silicon oil. The study is meaningful; however, some issues need to be addressed before its publication:

Answer:

Thank you very much for your positive feedback and kind words regarding our revisions. We are grateful for your continued support and thoughtful comments, which have significantly contributed to strengthening the manuscript. We will carefully address all the comments and make the necessary adjustments to ensure the paper meets the required standards. We appreciate your time and effort in reviewing the manuscript and look forward to submitting the final version with all corrections made.

We have modified the manuscript with a clearer expression and show the changes made in the revised paper in red color. The specific comments were answered point by point as follows.

Specific comments 1:

The write-up of this manuscript doesn’t sound scientifically, the authors should follow proper scientific writing methods.

Answer:

Thank you for your valuable feedback. We have carefully revised the entire manuscript to improve its scientific rigor and clarity. Colloquial or informal expressions have been replaced with academically appropriate phrasing, and the logical flow of the arguments has been significantly enhanced. Moreover, we have ensured consistent use of terminology, corrected grammatical issues, and employed passive voice and formal structures in line with scientific writing standards. We believe these revisions greatly improve the readability and professionalism of the manuscript. We hope these revisions will adequately address your comments.

Specific comments 2:

The title of the manuscript is not clear, consider revising it. The authors may consider a title such as “Research study on aging behaviours of insulating silicon oil for cable terminals based on chromatographic analysis” or improve it better.

Answer:

Thank you for your suggestion. We agree that the original title was not clear and we have revised it to:

“Research on Aging Behaviors of Insulating Silicone Oil for Cable Terminals Based on Chromatographic and Spectroscopic Analysis”

This new title better reflects the core aspects of the study, we hope this title will improve the clarity and precision of the manuscript.

Specific comments 3:

The manuscript has some grammar and typo mistakes that need to be improved or corrected. For example, 140℃, 350ppm, etc. There should be a space between a number and a unit like 140 ℃ and 350 ppm. Also, in some areas there is no space after a period or full stop like “discharges.Considering” (line 132), and “interference.Place” (line 134). Make the necessary corrections throughout your manuscript.

Answer:

We sincerely thank the reviewer for pointing out the grammar and typo issues in our manuscript. In response, we have carefully reviewed the entire manuscript and made the necessary corrections, including insert spaces between numbers and units and add spaces after periods and commas, et al. We sincerely apologize for the oversight and appreciate your attention to detail, which has helped improve the quality of our submission.

Revision:

Abstract

Line19-22

Accelerated thermal aging experiments (140 °C, 30 days) were conducted to simulate long-term aging of silicone oil. By integrating partial discharge, high-energy discharge, and breakdown experiments, the gas generation patterns of silicone oil under different stresses were systematically analyzed.

Line141-159

Oil chromatography analysis primarily detects seven gas components generated during aging degradation: H₂: Reflects moisture or hydrocarbon decomposition in insulation materials. CH₄, C₂H₆, C₂H₄, C₂H₂: Characterize hydrocarbon pyrolysis or corona discharge. CO, CO₂: Indicate decomposition of oxygen-containing groups in silicone oil. C₂H₂: A signature gas for high-energy discharges. Considering the instrument’s single-measurement requirement of 40 mL, extract 60 mL of post-discharge silicone oil sample. Then inject 1 mL of high-purity nitrogen as a carrier gas to assist gas separation and avoid oxygen interference. Place the sample in a constant-temperature shaker at 50 °C for 20 minutes to promote gas liberation. After shaking, use a gas-tight syringe to extract gas from the sample headspace, avoiding external air contamination, and immediately proceed to measurement. A chromatographic column selectively adsorbs gases based on polarity. Low-molecular-weight gases (H₂, CH₄) with weak adsorption elute first. Polar gases (CO, CO₂) interact strongly with the stationary phase, prolonging retention time. Hydrocarbon gases (C₂H₆, C₂H₄, C₂H₂) are separated sequentially by carbon chain length and double bond count. The thermal conductivity detector is used for detecting flameless combustion gases such as H₂, CO, and CO₂. Quantifies hydrocarbons (CH₄, C₂H₆) by ionizing them in a hydrogen flame and measuring the resulting current. Gas species are identified by comparing peak retention times with standard gas calibrations. Concentrations are calculated using the internal standard method, correlating peak areas with calibration curves.

Line213

Time (days) H₂ Concentration (ppm) ln⁡(C_max/(C_max-C))

5 80.77 0.12

10 150.2 0.32

30 323.3 0.87

Assuming C_max = 350ppm, The fitting result is k ≈ 0.029 day ⁻ ¹.

Specific comments 4:

The manuscript lacks consistency, for example, “Fig. 1.” and “Fig 2.” are used interchangeably. The authors should observe consistency throughout the manuscript.

Answer:

Thank you for highlighting this inconsistency. We have carefully revised the manuscript to ensure consistency in formatting throughout. All figure and table references have been updated to the standardized format “Fig X.” and “Table X.”. We ensured these changes are applied uniformly across the text to align with academic writing conventions. We sincerely apologize for the oversight and appreciate your attention to detail, which has helped improve the quality of our submission.

Specific comments 5:

In section 2 “Experimental and simulation methods”, which simulation method was used in this work and for what purpose? The ball and stick structure of silicon in Figure 1 is not clear, redraw and make it colourful.

Answer:

Thank you for your valuable comment. We apologize for the confusion caused by the section title. In fact, this work focuses solely on experimental investigations, and no simulation methods were employed. To avoid misunderstanding, we have revised the section title to “Experimental Methods” accordingly. Regarding Figure 1, we appreciate the reviewer’s suggestion. The original image has been redrawn using improved resolution and a clearer color scheme to enhance its visual clarity and readability. The updated figure is included in the revised manuscript.

Revision:

Specific comments 6:

The basic parameters given in Table 1 were they determined by the authors or other researchers? If authors, explain how they are determined or derived, and if other researchers, a reference is required.

Answer:

Thank you for this important clarification request. The parameters listed in Table 1 (e.g., density, kinematic viscosity, flash point, etc.) are not obtained experimentally by the authors but are sourced from the technical datasheet provided by the manufacturer, Wacker Chemie AG. We have updated the manuscript accordingly to state this explicitly.

Revision:

Line80-85

The silicone oil sample selected for testing was AK-50 insulating silicone oil produced by Wacker Chemie AG, Germany. Its primary component is polydimethylsiloxane (PDMS), featuring a linear chain structure composed of alternating silicon and oxygen atoms, with each silicon atom bonded to two methyl groups. The basic molecular structure is illustrated in Fig 1. The key physical and chemical properties of the silicone oil sample are summarized in Table 1(The technical datasheet provided by the manufacturer, Wacker Chemie AG).

Specific comments 7:

Why the aging characteristics of the insulating silicon oil are tested only at 140 ℃ instead of a range of temperatures?

Answer:

Thank you for your important comments. In this study, 140 ℃ was deliberately chosen as the aging temperature to represent an accelerated thermal aging condition, approximately 3 times higher than the typical operating temperature range (40-50 ℃) for insulating silicone oil in practical high voltage cable terminal. This approach allows for the efficient observation of thermal degradation and moisture-related changes within a manageable experimental timeframe. The suggestions from the reviewers are very correct. In our subsequent experimental plan, we will consider adding multiple heat aging temperature points for the experiments to study the effects of different heat aging temperatures on the insulating oil.

Specific comments 8:

Figures 2 to 7, Tables 2 to 5, and Equations 1 to 3 are not mentioned or cited anywhere in the text. Figures, Tables, and Equations should be part of the text.

Answer:

Thank you for carefully pointing out this important oversight. We have thoroughly revised the manuscript to ensure that all figures (Figs 2 to 7), tables (Tables 2 to 5), and equations (Equations 1 to 3) are clearly cited and discussed at appropriate points in the manuscript. These elements have now been fully integrated into the narrative to support and clarify the experimental results and analysis. We sincerely apologize for the omission and appreciate the reviewer’s detailed attention, which has helped us improve the manuscript's structure and readability.

Revision:

Line112-114

The silicone oil is sealed in a clean bottle and placed in a drying oven for high-temperature thermal aging. Fig 2 shows actual photographs of the oven and the silicone oil under different aging states.

Line126-127

The schematic diagram of the experimental circuit and the physical photograph of the needle-plate electrode are shown in Fig 3.

Line150-151

The schematic flow diagram illustrating the principle of oil chromatography detection is shown in Fig 4.

Line188-189

The oil chromatography data detected during continuous 30-day thermal aging are presented in Fig 5.

Line261-262

The infrared spectrum of the new silicone oil is presented in Fig 6, obtained from infrared spectroscopy analysis.

Line273-274

The results shown in Fig 7 indicate that no significant chemical bond breakage occurred in the silicone oil after 30 days of aging.

Line212

Taking H₂ as an example, The example calculation for H₂ is presented in Table 2:

Line224-225

The bond energies of other gases calculated using the aforementioned equations are presented in Table 3:

Line234-235

The oil chromatography data for different discharge stages are presented in Table 4.

Line249-250

The oil chromatography data after breakdown are presented in Table 5.

Line268-269

As can be seen from the figure above, the main absorption peaks of the silicone oil are listed in Table 6, along with the analysis of the corresponding functional groups.

Line206-207

Equation 1 and 2 are the first-order kinetic reaction formula, and equation 3 is Arrhenius formula.

Specific comments 9:

Why it is worthy reporting Figures 3 and 4 in the manuscript?

Answer:

Thank you for your question. Figures 3 and 4 serve important roles in illustrating the experimental setup. Figure 3 provides both a schematic diagram of the partial discharge (PD) test circuit and a photo of the needle-plate electrode, which are critical for understanding how discharge conditions were established and controlled. Figure 4 illustrates the configuration of the gas chromatography testing system, helping readers understand the sampling, degassing, and gas analysis procedures used for identifying degradation gases in silicone oil. These figures enhance the methodological clarity and reproducibility of our work and are therefore essential to retain in the manuscript.

Specific comments 10:

What is the unit for the y-axis in Figure 5? The y-axis title text direction should be changed.

Answer:

Thank you for bringing up this matter. We have revised Figure 5 accordingly and added unit annotations.

We hope this revision satisfies your concerns.

Revision:

Line183

Line185

Specific comments 11:

Figures resolution is not good, consider redrawing them and make attractive.

Answer:

Thank you for pointing this out. We have reviewed all figures and are in the process of redrawing them with enhanced clarity and consistency. All revised figures are exported at high resolution with improved visual contrast, uniform font styling, and clear annotations. These updates ensure professional presentation quality and facilitate better interpretation by readers.

We hope this revision satisfies your concerns.

Revision:

Line183

Line185

Line266

Line271

Specific comments 12:

Equations 1 to 3 are confusing; they should be properly written.

Answer:

Thank you for this valuable comment. We have revised Equations (1) to (3) to adopt a clear and consistent scientific format. Each equation is now properly centered, numbered.

We hope this revision satisfies your concerns.

Revision:

dC/dt=k×(C_max-C) (1)

ln⁡(C_max/(C_max-C))=kt (2)

k=A×e^(-E_a/RT) (3)

Specific comments 13:

Generally, the manuscript in its current version lacks sufficient novelty. The author(s) should revise or reformulate the abstract, introduction, methodology, and conclusion to clearly show the strength and novelty of the manuscript.

Answer:

We thank the reviewer for this important suggestion. In response, we have substantially revised the abstract, introduction, methodology, and conclusion to highlight the scientific contributions and novelty of our work. The major novelties are now clearly emphasized as follows:

This is the first study to propose gas ratio-based fault diagnosis criteria specifically for insulating silicone oil, which are not covered by existing standards developed for transformer oil. We introduce quantitative gas ratios (e.g., H₂/CH₄, C₂H₄/C₂H₆, C₂H₂/C₂H₄) as effective indicators of fault types in silicone oil systems. We further enhance the methodology by combining accelerated thermal aging experiments with first-order kinetic modeling and Arrhenius analysis to investigate the chemical origins of gas formation. These findings fill a practical gap in diagnostic approaches for silicone oil-insulated cable terminals and contribute to future predictive maintenance practices. We hope that the revised manuscript more effectively communicates the novelty and significance of our research.

We hope this revision satisfies your concerns.

Revision:

Abstract

With the widespread deployment of high-voltage cable terminals in power systems, insulating silicone oil has become a critical medium due to its superior dielectric and thermal properties. However, conventional diagnostic methods such as the three-ratio gas analysis developed for transformer oil have proven ineffective for silicone oil, owing to its distinct chemical structure and degradation behavior. To address this, this study aims to establish a fault-type identification method specifically for silicone oil to enhance the operational reliability of cable terminals. Accelerated thermal aging experiments (140 °C, 30 days) were conducted to simulate long-term aging of silicone oil. By integrating partial discharge, high-energy discharge, and breakdown experiments, the gas generation patterns of silicone oil under different stresses were systematically analyzed. Gas chromatography (GC) and infrared spectroscopy (IR) were employed to track gas composition and chemical structural changes. The results propose the following diagnostic criteria: H₂/

---

## [Decision Letter · Decision Letter 1]

6 Aug 2025

Dear Dr. zhang,

Thank you for submitting your manuscript to PLOS ONE. After careful consideration, we feel that it has merit but does not fully meet PLOS ONE’s publication criteria as it currently stands. Therefore, we invite you to submit a revised version of the manuscript that addresses the points raised during the review process.

We look forward to receiving your revised manuscript.

Kind regards,

Antonio Javier Nakhal Akel, PhD

Academic Editor

PLOS ONE

Journal Requirements:

Additional Editor Comments :

Dear Authors,

Please find below the comments and suggestions provided by the reviewers.

Should you require any further information or clarification, please do not hesitate to contact me.

Best regards,

Antonio Nakhal

Academic Editor

Reviewers' comments:

Reviewer's Responses to Questions

**Comments to the Author**

Reviewer #1: All comments have been addressed

Reviewer #2: (No Response)

2. Is the manuscript technically sound, and do the data support the conclusions?

Reviewer #1: Yes

Reviewer #2: Yes

3. Has the statistical analysis been performed appropriately and rigorously?

Reviewer #1: No

Reviewer #2: Yes

4. Have the authors made all data underlying the findings in their manuscript fully available?

Reviewer #1: Yes

Reviewer #2: Yes

5. Is the manuscript presented in an intelligible fashion and written in standard English?

Reviewer #1: Yes

Reviewer #2: Yes

Reviewer #1: Some improvements have been made; however, there are still some areas which needs further improvements.

1) The author should define the colours used in Figure 1, for example red colour represent Oxygen atom etc. Use the standard colour for each atom.

2) I recommend Figure 1 caption to read “Chemical structure of polydimethylsiloxane as the primary component in AK-50 insulating silicone oil”.

3) There are still some typo errors for example, no space between a number and a unit in 250℃ and 55℃ (Table 1). Make the necessary corrections throughout your manuscript.

4) I recommend Figure 2 caption to read “Pictorial representation of (a) oven and (b) silicone oil at different aging stages”.

5) I recommend Figure 3 caption to read “Schematic diagram of (a) partial discharge experimental circuit and (b) physical drawing of needle plate electrode”.

6) I recommend Figure 3 caption to read “Oil chromatography data for continuous aging over 30 days for (a) CO₂, CO, and H₂; and (b) C₂H₆, C₂H₄, and CH₄”.

7) Use the correct unit for wavenumber (cm⁻¹) in Figures 6 and 7, apply superscript and use the correct minus sign from symbols.

8) Figures 6 and 7 can be drawn in the same graph; how can the values of Transmittance (Figure 7) be negative?

Reviewer #2: The authors have addressed all the comments raised. Therefore, the manuscript is suitable for publication pending minor revisions. The authors should ensure that the manuscript adheres to the journal's guidelines and that all figures are presented in high resolution.

**Do you want your identity to be public for this peer review?** For information about this choice, including consent withdrawal, please see our Privacy Policy

Reviewer #1: No

Reviewer #2: **Yes: ** Henry Kahimbi

---

## [Author Response · Author response to Decision Letter 2]

11 Aug 2025

Response to Reviewer #1:

General comments:

Some improvements have been made; however, there are still some areas which needs further improvements.

Answer:

Thank you very much for your positive feedback and kind words regarding our revisions. We are grateful for your continued support and thoughtful comments, which have significantly contributed to strengthening the manuscript. We will carefully address all the comments and make the necessary adjustments to ensure the paper meets the required standards. We appreciate your time and effort in reviewing the manuscript and look forward to submitting the final version with all corrections made.

We have modified the manuscript with a clearer expression and show the changes made in the revised paper in red color. The specific comments were answered point by point as follows.

Specific comments 1:

The author should define the colours used in Figure 1, for example red colour represent Oxygen atom etc. Use the standard colour for each atom.

Answer:

Thank you for the valuable suggestion. The colours used in Figure 1 have now been defined in the figure caption. We have also ensured that standard colour conventions are followed(e.g., red for oxygen, white for hydrogen, grey for carbon, etc.).

Revision:

Fig 1. Chemical structure of polydimethylsiloxane as the primary component in AK-50 insulating silicone oil

Specific comments 2:

I recommend Figure 1 caption to read “Chemical structure of polydimethylsiloxane as the primary component in AK-50 insulating silicone oil”.

Answer:

Thank you for the helpful suggestion. We have revised the caption of Figure 1 to read: “Chemical structure of polydimethylsiloxane as the primary component in AK-50 insulating silicone oil’’ as recommended.

Revision:

Fig 1. Chemical structure of polydimethylsiloxane as the primary component in AK-50 insulating silicone oil

Specific comments 3:

There are still some typo errors for example, no space between a number and a unit in 250℃ and 55℃ (Table 1). Make the necessary corrections throughout your manuscript.

Answer:

Thank you for pointing this out. We have carefully checked the manuscript and corrected all instances of missing spaces between numbers and units (e.g., “250℃” changed to “250 ℃”) in Table 1 and throughout the manuscript as necessary.

Revision:

Table 1. Basic parameters of test sample silicone oil.

Basic parameters Performance index

Density (25 ℃) 0.96 g/cm3

Kinematic Viscosity 50 mm2 s-1

Coefficient of thermal expansion at 0-150 ℃ 9.5 cm3/ (cm3 ℃ )×10-4

Thermal conductivity at 50 ℃ 0.15 W K-1 m-1

Flash point >250 ℃

Pour point 55 ℃

Volatility <2%

Specific comments 4:

I recommend Figure 2 caption to read “Pictorial representation of (a) oven and (b) silicone oil at different aging stages”.

Answer:

Thank you for the helpful suggestion. We have revised the caption of Figure 2 to read: “Pictorial representation of (a) oven and (b) silicone oil at different aging stages”as recommended.

Revision:

(a) oven (b) silicone oil at different aging stages

Fig 2. Pictorial representation of (a) oven and (b) silicone oil at different aging stages

Specific comments 5:

I recommend Figure 3 caption to read “Schematic diagram of (a) partial discharge experimental circuit and (b) physical drawing of needle plate electrode”.

Answer:

Thank you for the insightful suggestion. We have revised the caption of Figure 3 to read: “Schematic diagram of (a) partial discharge experimental circuit and (b) physical drawing of needle plate electrode”as recommended.

Revision:

(a) partial discharge experimental circuit (b) physical drawing of needle plate electrode

Fig 3. Schematic diagram of (a) partial discharge experimental circuit and (b) physical drawing of needle plate electrode

Specific comments 6:

I recommend Figure 3 caption to read “Oil chromatography data for continuous aging over 30 days for (a) CO₂, CO, and H₂; and (b) C₂H₆, C₂H₄, and CH₄”.

Answer:

Thank you for the helpful suggestion. We have revised the caption of Figure 3 for clarity and consistency with the manuscript. It now reads: “Oil chromatography results over 30 days of aging: (a) CO₂, CO, and H₂; (b) C₂H₆, C₂H₄, and CH₄”.

Revision:

(a) CO₂, CO, and H₂

(b) C₂H₆, C₂H₄, and CH₄.

Fig 5. Oil chromatography data for continuous aging over 30 days for (a) CO₂, CO, and H₂; and (b) C₂H₆, C₂H₄, and CH₄

Specific comments 7:

Use the correct unit for wavenumber (cm⁻¹) in Figures 6 and 7, apply superscript and use the correct minus sign from symbols.

Answer:

Thank you for pointing this out. We have corrected the unit for wavenumber in Figures 6 and 7 to “cm⁻¹” using the appropriate superscript format and the correct minus sign symbol. These changes have been applied consistently throughout the figures and captions.

Revision:

Fig 6. Infrared Spectrum of AK50 Silicone Oil and Aging Data Over 30 Days.

Specific comments 8:

Figures 6 and 7 can be drawn in the same graph; how can the values of Transmittance (Figure 7) be negative?

Answer:

Thank you for your valuable comment. In the revised manuscript, Figures 6 and 7 have been combined into a single graph for better comparison, as suggested. Regarding the negative transmittance values in the original Figure 7, we would like to clarify that transmittance itself cannot be negative; the apparent negative values were due to a vertical offset applied during plotting for visual clarity. This issue has been corrected in the revised figure to avoid any confusion.

Revision:

Fig 6. Infrared Spectrum of AK50 Silicone Oil and Aging Data Over 30 Days.

---

## [Decision Letter · Decision Letter 2]

29 Sep 2025

Research on Aging Behaviors of Insulating Silicone Oil for Cable Terminals Based on Chromatographic and Spectroscopic Analysis

PONE-D-25-13265R2

Dear Dr. zhang,

We’re pleased to inform you that your manuscript has been judged scientifically suitable for publication and will be formally accepted for publication once it meets all outstanding technical requirements.

Kind regards,

Antonio Javier Nakhal Akel, PhD

Academic Editor

PLOS ONE

Additional Editor Comments (optional):

Reviewers' comments:

Reviewer's Responses to Questions

**Comments to the Author**

Reviewer #2: All comments have been addressed

Reviewer #3: All comments have been addressed

2. Is the manuscript technically sound, and do the data support the conclusions?

Reviewer #2: Yes

Reviewer #3: Yes

3. Has the statistical analysis been performed appropriately and rigorously?

Reviewer #2: Yes

Reviewer #3: Yes

4. Have the authors made all data underlying the findings in their manuscript fully available?

Reviewer #2: Yes

Reviewer #3: Yes

5. Is the manuscript presented in an intelligible fashion and written in standard English?

Reviewer #2: Yes

Reviewer #3: Yes

Reviewer #2: (No Response)

Reviewer #3: I appreciate the efforts of all the authors; you have addressed all the reviewers’ questions. You may now receive the final decision from the editor’s office.

**Do you want your identity to be public for this peer review?** For information about this choice, including consent withdrawal, please see our Privacy Policy

Reviewer #2: **Yes: ** Henry Kahimbi

Reviewer #3: No

---

## [Editor Report · Acceptance letter]

PONE-D-25-13265R2

PLOS ONE

Dear Dr. zhang,

I'm pleased to inform you that your manuscript has been deemed suitable for publication in PLOS ONE. Congratulations! Your manuscript is now being handed over to our production team.

Kind regards,

on behalf of

Dr. Antonio Javier Nakhal Akel

Academic Editor

PLOS ONE